# Structure Investigation of Titanium Metallization Coating Deposited onto AlN Ceramics Substrate by Means of Friction Surfacing Process

**Tomasz Chmielewski ***[ID], **Michał Hudycz, Arkadiusz Krajewski, Tadeusz Sałaciński**[ID], **Beata Skowrońska and Rafał Świercz**[ID]

Warsaw University of Technology, 02-524 Warsaw, Poland; m.hudycz@wip.pw.edu.pl (M.H.); a.krajewski@wip.pw.edu.pl (A.K.); t.salacinski@wip.pw.edu.pl (T.S.); b.skowronska@wip.pw.edu.pl (B.S.); rsw@meil.pw.edu.pl (R.Ś.)

**\*** Correspondence: t.chmielewski@wip.pw.edu.pl; Tel.: +48-22-849-9797

**Abstract:** The article presents selected properties of a titanium metallization coating deposited on aluminum nitride (AlN) ceramics surface by means of the friction surfacing method. Its mechanism is based on the formation of a joint between the surface of an AlN ceramics substrate and a thin Ti coating, involving a kinetic energy of friction, which is directly converted into heat and delivered in a precisely defined quantity to the resulting joint. The largest effects on the final properties of the obtained coating include the high affinity of titanium for oxygen and nitrogen and a relatively high temperature for the deposition process. The titanium metallization coating was characterized in terms of surface stereometric structure, thickness, surface morphology, metallographic microstructural properties, and phase structure. The titanium coating has a thickness ranging from 3 to 7 μm. The phase structure of the coating surface (XPS investigated) is dominated by $TiN_xO_y$ with the presence of $TiO_x$, TiN, metallic Ti, and AlN. The phase structure deeper below the surface (XRD investigated) is dominated by metallic Ti with additional AlN particles originating from the ceramic substrate due to friction by titanium tools.

**Keywords:** titanium coating; friction surfacing; metallization of AlN ceramics; phase structure

---

## 1. Introduction

In recent years, the use of aluminum nitride (AlN) ceramics in the many industry branches (chemical, electronics, energy, petrochemical) has been increasing and often requires the production of permanent ceramic–metal joints [1–5]. Bonding ceramic materials with metals is one of the most difficult tasks in joining engineering. Difficulties in bonding the presented pair of materials result from the extremely different physical and chemical properties of ceramic and metal materials [6–8]. One of the major problems with the mismatch of ceramics and metal from the point of view of a creation of the joint is primarily the difference in the structure of atomic, ionic, and mixed bonds, and the insufficient wettability of the ceramic surface for most of the liquid metals. Different values of the thermal expansion coefficient a (AlN-5.6 × 10$^{-6}$ K$^{-1}$; Ti-8.6 × 10$^{-6}$ K$^{-1}$) and the thermal conductivity coefficient λ (AlN-140 ÷ 180 W/m·K; Ti-22,4 W/m·K) are equally important for the properties of the formed joint [9–12]. It is due to the relatively high level of internal stresses resulting, for example, from the technological conditions of coating application and extremely different physical properties of ceramics and metals.

In most cases of bonding ceramics with metals, during the joining process, the ceramics remain in the solid state [13–17], and the metal joined to it is very often in a liquid state. The metallization

coating described in this article has been made using the friction method, which allows the creation of a joint between the ceramic substrate and the metallization coating, where during the joining process, both materials remain in a solid state. Currently, there is an interest in various types of surface modification processes and bonding with a mechanical method of energy supply; [18–20] investigated this approach and found that it makes bonding easier, even for extremely different materials. In the case described in this article, the maintenance of the metal of the resulting coating in the solid state (characterized by high plasticity) results from the supply of energy to the bonding area on the mechanical way (by friction). This method of heating enables precise control of the amount of energy supplied and the narrowing of the place of heating only to the area close to the friction surface of the forming joint [21–25].

The purpose of producing a thin titanium coating is the metallization of the ceramic surface, which among others modifies the chemical properties of the surface and facilitates, for example, the soldering of ceramic products with metal through the standard titanium coating. Titanium as a coating material was used due to its significant chemical affinity for the basic components of ceramics, including aluminum and nitrogen [26–30].

The purpose of this work is to characterize the surface and phase structure of the Ti metallization coating deposited on AlN ceramic surface by means of the friction surfacing method.

## 2. Materials and Methods

In this study, high purity (99%) AlN ceramics manufactured at the Institute of Electronic Materials Technology (Warsaw, Poland) and titanium (standard Grade 2) were used for the research.

The titanium metallization coating was deposited on the ceramic surface by rotational friction with the front of a cylindrical titanium tool (indicated by the arrow in Figure 1a) with an external diameter of 9 mm and an opening of 3 mm in the axis. The rotating tool additionally made a linear or spiral movement to obtain a coating surface much larger than the tool's working surface (Figure 1b). The described tool has been mounted in a holder, enabling the transfer of torque and controlling the pressure on the friction surface. The mechanism of coating formation consists of the plastic galling of the tool's face material on the ceramic surface. The literature of the subject indicates that during the bonding process in the solid state, it is possible to obtain high quality enough and permanent joints of materials with different properties [31–33]. In order to avoid the oxidation of titanium, the working area (Figure 1b) was surrounded by a cylindrical barrier inside of which was supplied with 99.995% of argon shielding gas with a flow rate of 5 l/min, providing oxygen concentration below 5 ppm.

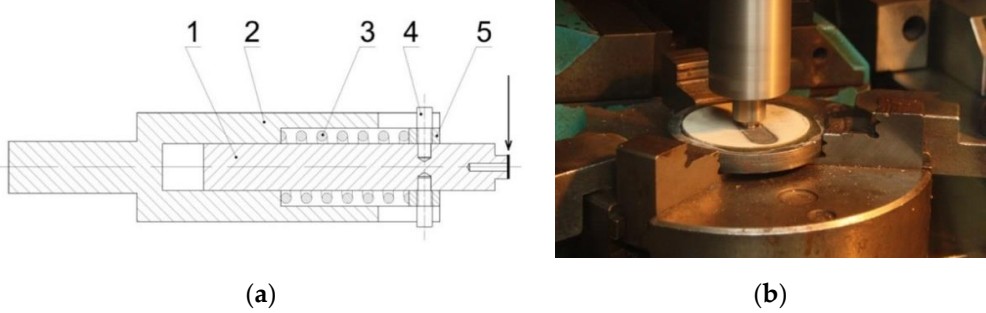

(**a**)   (**b**)

**Figure 1.** Friction tool: (**a**) schema of Ti friction tool in holder: (1) friction tool, (2) holder, (3) spring, (4) and (5) spring tension adjustment items; (**b**) the friction tool during operation and a view of the emerging coating.

The friction metallization process was carried out at the Cincinnati Arrow 500 numerical machining center (Cincinnati Machine, Birmingham, U.K.), the installed substrate and tool during the coating forming process is shown in Figure 1b. Figure 2 shows a view of a coating consisting of several string beads with a 3 mm overlap, forming a larger surface.

Measurements of the stereometric structure of the coating's surface layer were carried out on the optical µscan select profilometer from Nanofocus (Oberhausen, Germany). SEM images have been prepared with a BSE detector on microscop JEOL (Peabody, MA, USA). A linear and surface distribution of elements in the cross-section of the substrate and coating system has been prepared by electronomicroscopic examinations using the Talos F200X (Waltham, MA, USA) transmission microscope in EDX mode. The XRD phase analysis of the coating has been realized with X-ray diffractometer SmartLab 3kW Rigaku (Tokyo, Japan). The photoelectron spectroscopy (XPS) have been carried out on the apparatus AXIS SUPRA (ESCA) XPS Kratos (Wharfside, Manchester, U.K.).

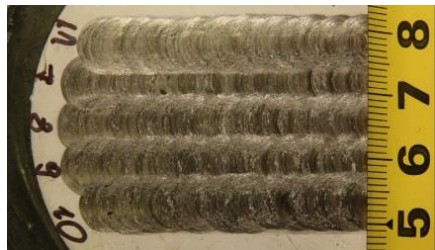

**Figure 2.** Coating consisting of several string beads with overlapping.

## 3. Results of Structure Investigation

### 3.1. Stereometric Structure of the Surface Layer of the Coating

From the point of view of the purpose of the subject coating as an intermediate layer in the joining processes of metallized ceramics with metals, an important feature in addition to e.g., braze wettability is also the stereometric structure of the surface, which has a significant effect on solderability. The parameters of the measurements that were carried out were a measuring field (x direction) 5 mm × (y direction) 4.5 mm. Parameters of the surface roughness after filtration were determined as Gauss 0.08 mm, and a stereometric map of the examined surfaces was made, which is shown in Figure 3a. The surface structure of the tested coatings applied mechanically is anisotropic; it shows a clear directionality of elevations and dimples, which is a representation of the tool's rotational motion with simultaneous feed. A local imperfection was also noted on the surface of the coating (blue spot in Figure 3). Figure 4 presents graphs showing the cloud of registered roughness profiles in the x and y directions. Individual profiles for both directions obtained in the middle of the tested field were also shown, as well as the average value of $R_a$ and $R_z$ for all measurements in both directions. The roughness parameters of the surface are on the relative low level from the used deposition method point of view and do not require further machining for possible surface brazing. The mean values are X direction, $R_a$-0.404 µm, $R_z$-1.95 µm; Y direction, $R_a$-0.909 µm, $R_z$-6.72 µm.

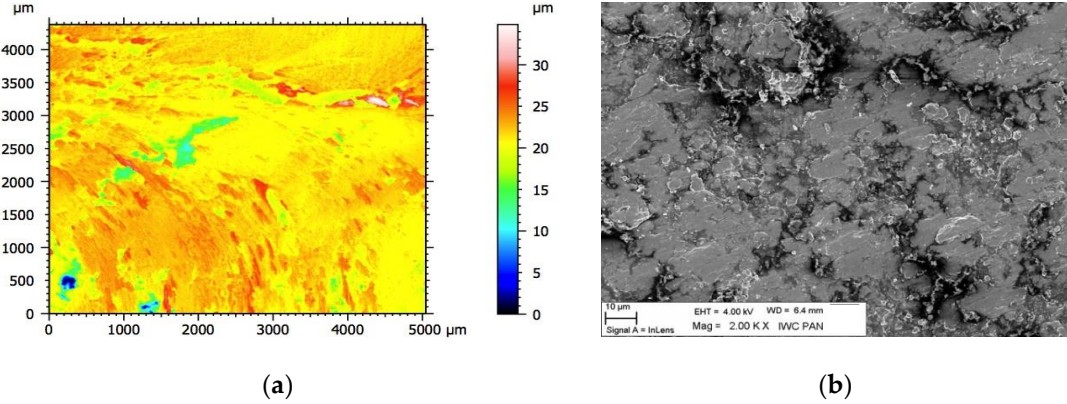

(**a**)                                   (**b**)

**Figure 3.** View of coating surface: (**a**) Stereometric map of the metallization coating surface; (**b**) SEM view of the coating surface.

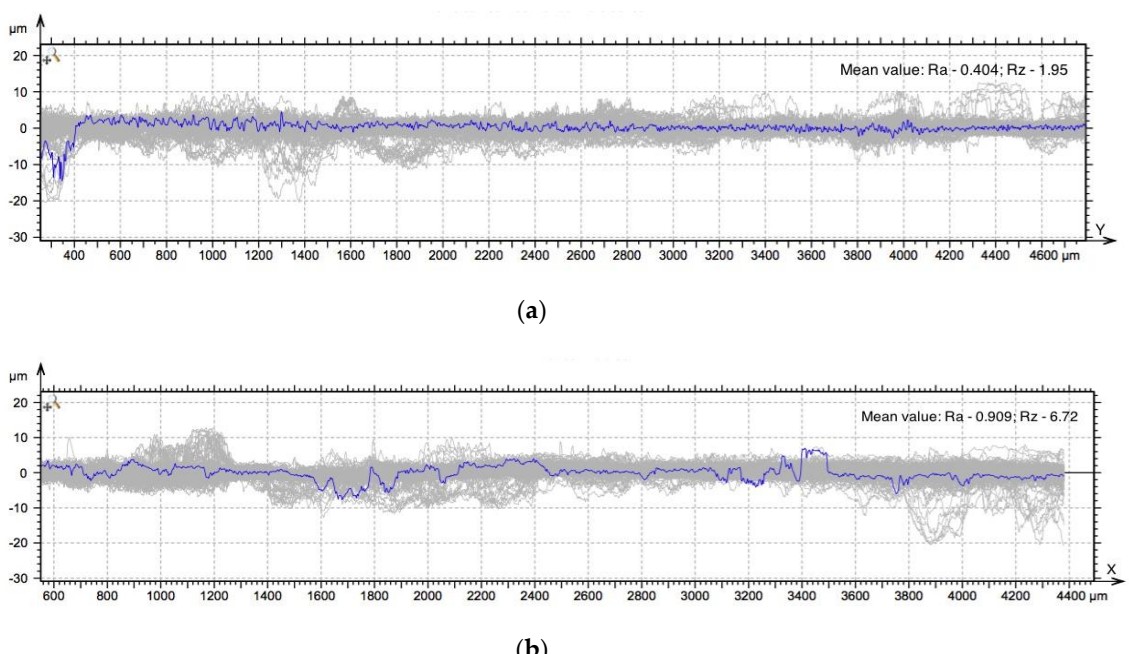

(a)

(b)

**Figure 4.** Roughness graphs and profiles in direction: (**a**)–x; (**b**)–y.

## 3.2. Microstructure of the Coating

Figure 5a shows the morphology of the breakthrough surface of ceramic substrate joint (AlN) with a metallic coating (Ti). SEM images created with a BSE detector show a mass contrast; observations were carried out on the surface of the breakthrough obtained from the simple three-point bending test of the sample (stretching on the coating side), which is demanding and also consists of a difficult qualitative test of the adhesion between coating and substrate. No chipping was observed in the joint area.

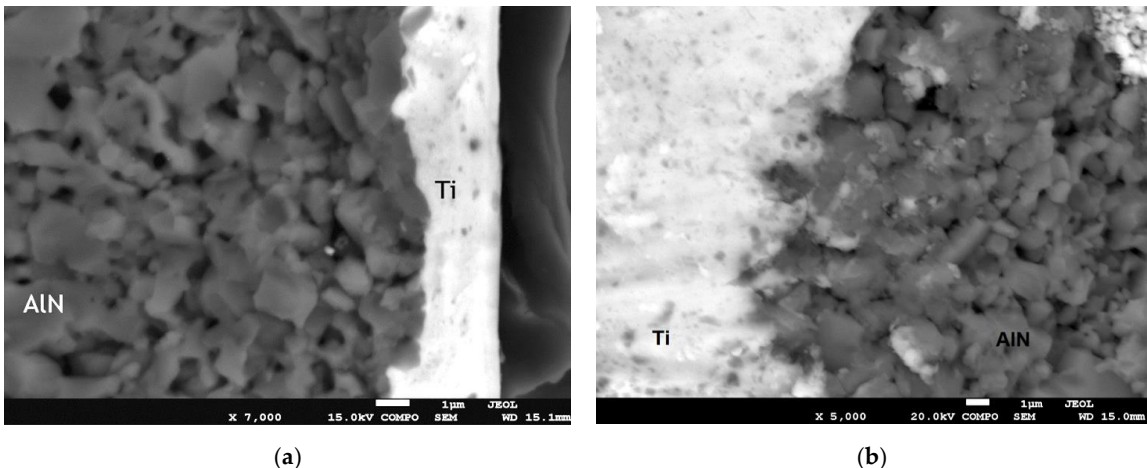

(a)

(b)

**Figure 5.** SEM with BSE detector view of coating: (**a**) cross-section through substrate AlN-coating Ti system; (**b**) view of the surface side on the border of the titanium metallization coating and the uncoated substrate.

The coating material was applied very well without discontinuities; it connected to the substrate and effectively filled the unevenness of the ceramic surface. The structure of the substrate shows the characteristics of AlN ceramics being a sinter of separate grains with characteristic porosity. No cracks in ceramics caused by frictional metallization process were observed. The coating thickness ranges from about 3 to 7 μm. The titanium coating tightly covers the ceramic surface, and its volume shows

the inclusions of the submicrometric ceramic grains coming from the friction surface, the presence and distribution of which in the coating are stochastic and additionally confirm a high degree of plasticization of the titanium grains during coating production. Figure 3b shows the SEM image of the coating from the surface view at its boundary with the substrate. In all types of coatings, the coverage boundary zone is a sensitive area with a high risk of delamination, which was investigated by [33–37]. It is due to the relatively high level of internal stresses resulting, for example, from the technological conditions of coating application and extremely different physical properties of ceramics and metals, such as the thermal expansion coefficient and thermal conductivity coefficient [38–41]. In the case described, no tendency for delamination of the coating was observed. The boundary line of the coating is developed, but without cracks and discontinuities.

The presence of aluminum (AlN ceramic grains) in the coating is also confirmed by the surface and linear distributions of elements on the cross-sectional surface of the substrate-coating system, which is presented in Figures 6 and 7. Linear distributions of elements have been divided into two parts: the first (Figure 6) contains N, Al, and Ti; the second (Figure 7) contains O, Al, and Ti. The separation results from the fact that the N and O lines are close together and deconvolution had to be applied to calculate them correctly.

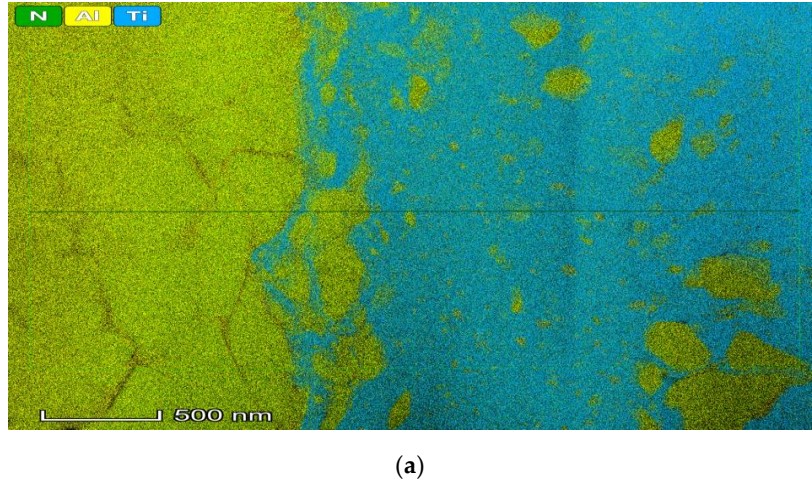

(**a**)

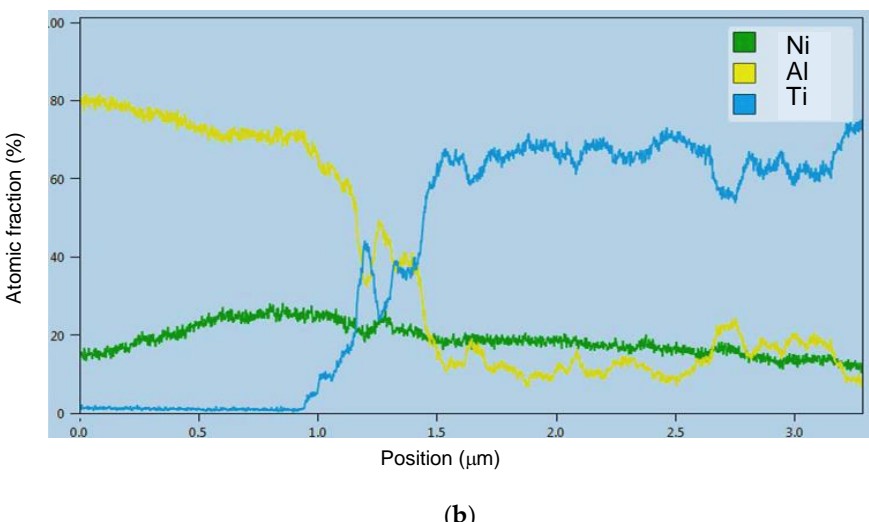

(**b**)

**Figure 6.** EDX view of interface area of coating and substrate with: (**a**) surface distribution of N, Al, and Ti; (**b**) Linear distribution of elements on the surface of the cross-section through the substrate and coating system.

The nature of changes in the concentration of titanium and aluminum at the AlN–Ti interface does not indicate the presence of the wide diffusion transition zone that occurs in ceramic–metal joints obtained by high-temperature sintering/vacuum bright annealing and other high-energy metallization methods [15,39].

The presence of inclusions rich in Al of the size of several hundred nanometers in the part of the Ti coating is visible. Their distribution is random, and their size is in the range from a few to about 500 nanometers. A profile with no sharp corners is favorable from the point of view of residual stress distribution. There are no free spaces between the Ti matrix and ceramics inclusions; however, short, single, randomly located cracks in ceramics were observed. The presence of oxygen in the Ti layer has also been noted. Oxygen and nitrogen are present in the Ti layer. There is little oxygen inside AlN grains; most of it is present at the grain edges.

Figure 7 shows the concentration of oxygen in the area of the ceramic grain boundaries, the origin of which most likely results from the technological conditions of sintering AlN ceramics. Oxidation of the AlN ceramic grain boundaries was recorded in the entire volume of the tested substrate. The spectra in Figure 7b from the areas marked with colored rectangles in Figure 7a correspond to the color of the markings. The aluminum in the Ti layer indicated in Figure 6a is most likely not in the free form, but in compounds with nitrogen and oxygen. There was no diffusion of Ti into the substrate.

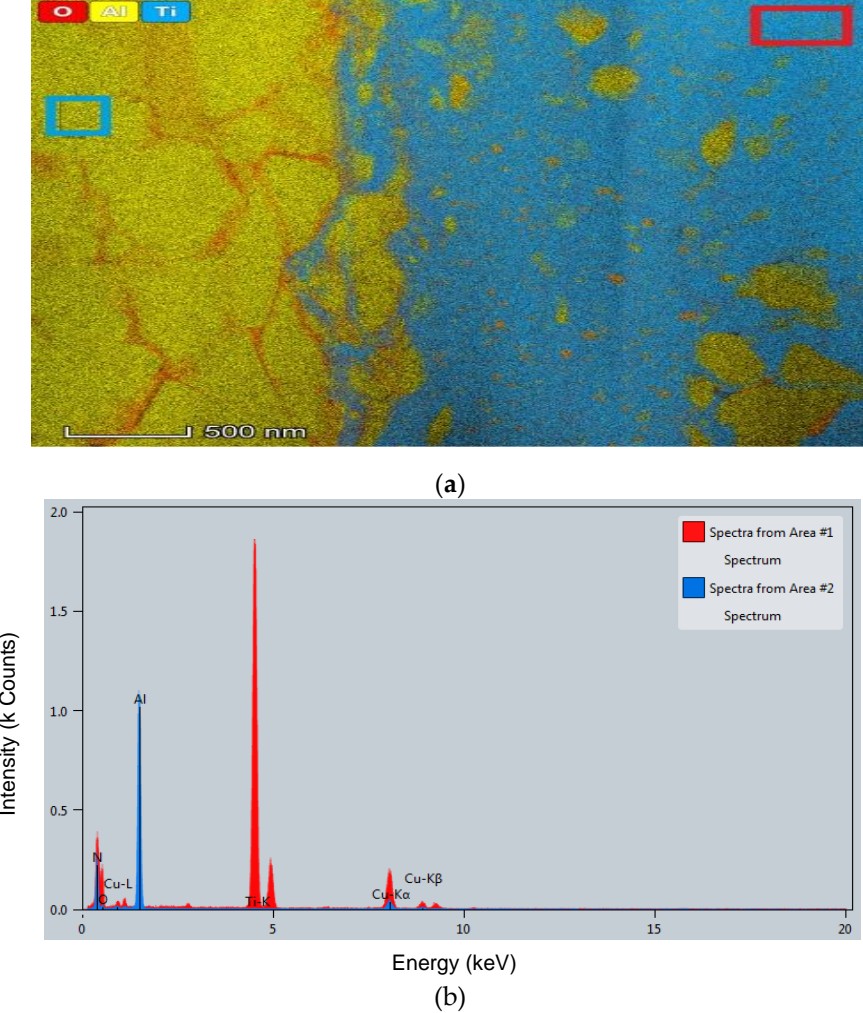

(**a**)

(b)

**Figure 7.** EDX view of interface area of coating and substrate with: (**a**) surface distribution of O, Al, and Ti; (**b**) The spectral analysis from the areas marked with rectangles correspond to the color of the spectrum.

### 3.3. Phase Analysis of the Titanium Metallization Coating

The phase structure of the Ti-based metallization coating is very important due to the required surface solderability if the coating will be used for further joining with metals [42,43]. The problem stems from the difference in the relatively good wettability of pure titanium and the limited wetting of oxides and titanium nitrides with most solders for brazing [44].

The XRD phase analysis (Figure 8) of the deposited coating showed a fairly high homogeneity of the metallic structure of the matrix, consisting mostly of pure titanium, which is practically amorphous (noticeably blurred diffraction signal). The interaction of titanium with air components was limited by using argon protection during the coating process. XRD analysis does not confirm a significant share of Ti chemical solutions that are very likely to occur, nor does XRD analysis confirm a significant share (above sensitivity of this method) of other titanium chemicals. The approximate share by volume of the crystal phase in the structure, identified with the XRD method, is approximately 5%. Additionally, no signal was recorded from the potential diffusion transitional layer. On the XRD diffraction pattern, one can observe a signal from titanium, AlN (ceramic substrate and particles obliterated in the coating), and the $Al_8O_3N_6$ compound being a component of the ceramic substrate.

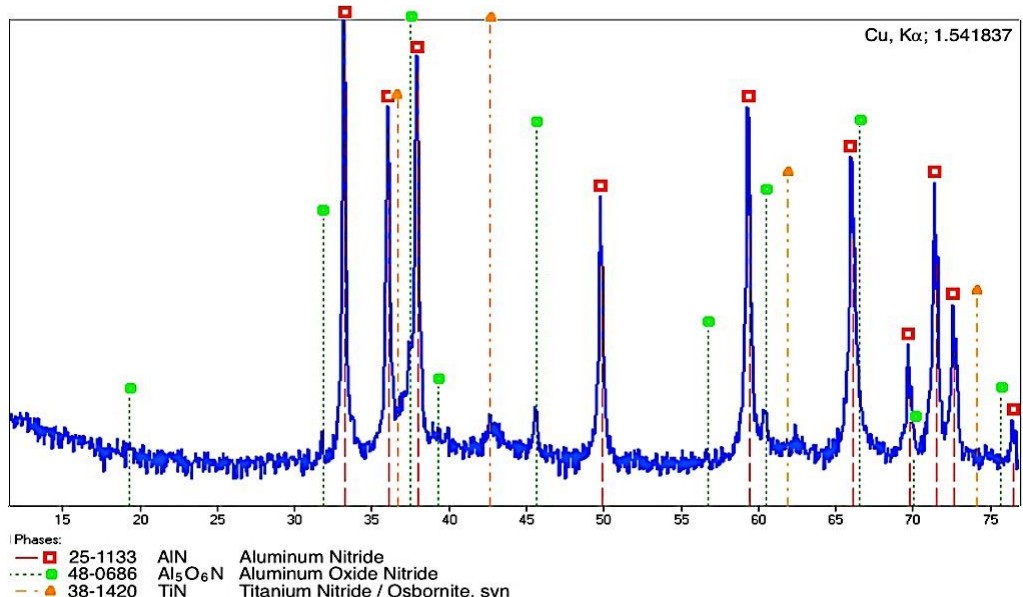

| Phases: | | | |
|---|---|---|---|
| ▢ | 25-1133 | AlN | Aluminum Nitride |
| ● | 48-0686 | $Al_5O_6N$ | Aluminum Oxide Nitride |
| ▲ | 38-1420 | TiN | Titanium Nitride / Osbornite, syn |

**Figure 8.** XRD diffractogram of Ti coating deposited onto aluminum nitride (AlN) ceramics substrate.

In the next step, X-ray photoelectron spectroscopy (XPS) has been carried out. The tests were performed on the apparatus AXIS SUPRA (ESCA) XPS Kratos. The measurements used a 225 W monochrome Al source. The surface charge accumulating on the sample was neutralized by (flood gun) an electron bombardment of the surface. The spectra obtained were calibrated to signal C 1*s* = 284.8 eV. The XPS review spectrum (Figure 9) shows the photoelectron and Auger electron signals for the given elements on the atomic bond energy scale. In the XPS technique, the signal is obtained from a thickness of 3 ÷ 15 nm (the technique is surface sensitive); hence, it refers to the surface layer of the coating material. A high-resolution spectrum of the Ti 2*p* region has been presented in Figure 10.

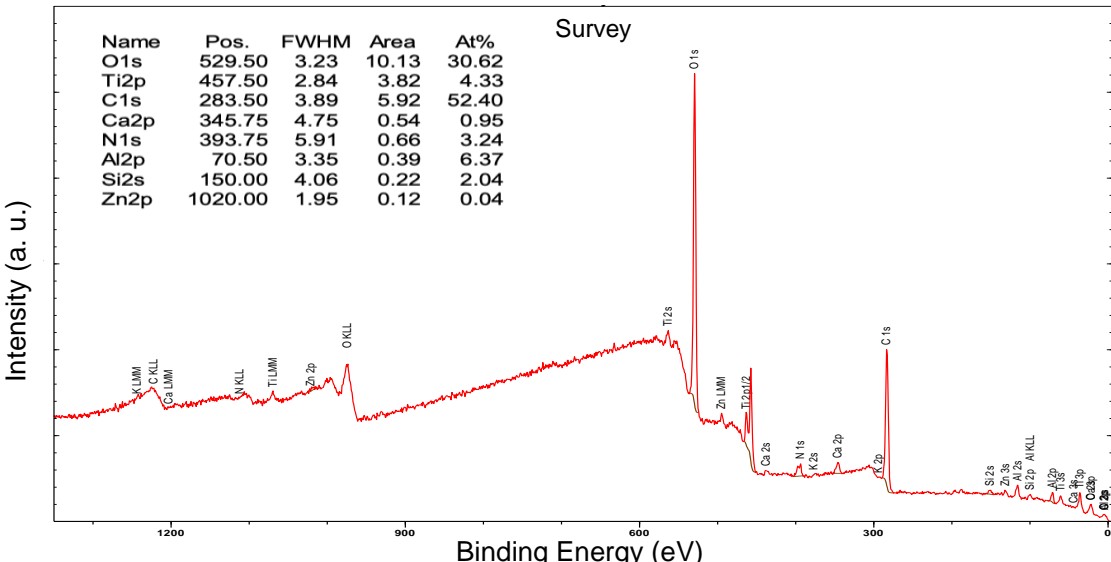

**Figure 9.** Review spectrum of the coating's surface.

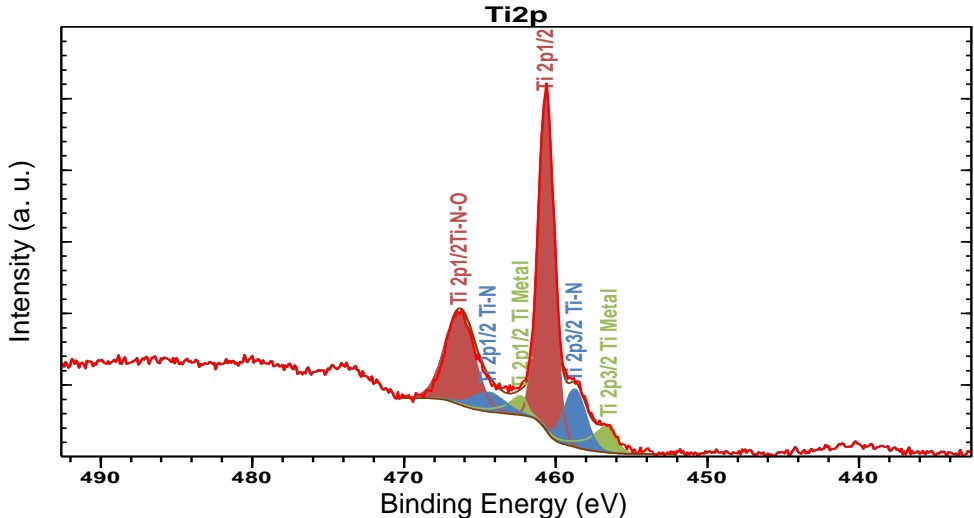

**Figure 10.** High resolution spectrum of the Ti 2*p* region.

Low amounts of carbon impurities and trace amounts of potassium were detected on the surface of deposited Ti coating. Individual components were assigned based on the literature data, respectively of atomic bonds (Table 1). The dominant form of titanium on the sample surface is most likely $TiN_xO_y$. The signal at this binding energy can also be attributed to $TiO_x$, where x equals approximately 2. The results of the XPS investigation indicate the presence of TiN, $TiO_xN_y$ compounds, metallic Ti, AlN, and $AlO_xN_y$ compounds.

**Table 1.** Summary of compounds identified on the surface of the coating: organic compounds (C 1*s*; K 2*p*); Ti compounds (Ti 2*p*); O compounds (O 1*s*); N compounds (N 1*s*); Al compounds (Al 2*p*).

| Region | Name | Position | Line Shape | FWHM | Raw Area | R.S.F. | %At. Conc * |
|---|---|---|---|---|---|---|---|
| C 1*s* | C–C, C–H | 284.50 | GL (30) | 1.71 | 4386.44 | | 42.03 |
| | C–OH, C–O–C | 286.50 | GL (30) | 1.60 | 5072.92 | | 48.53 |
| | C=O | 288.00 | GL (30) | 1.60 | 784.385 | 1 | 7.50 |
| | O–C=O | 289.50 | GL (30) | 1.60 | 103.182 | | 0.98 |
| | carbonates | 292.85 | GL (30) | 0.77 | 100.548 | | 0.96 |
| K 2*p* | K 2$p_{3/2}$ | 295.14 | GL (30) | 1.44 | 208.884 | 3.97 | 100 |
| | K 2$p_{1/2}$ | 297.44 | GL (30) | 1.71 | 104.442 | | |
| Ti 2*p* | Ti 2$p_{1/2}$ Ti–N–O | 460.63 | GL (30) | 1.22 | 4797.62 | | 71.95 |
| | Ti 2$p_{1/2}$ Ti–N–O | 466.33 | GL (30) | 2.23 | 2398.81 | | |
| | Ti 2$p_{3/2}$ Ti Metal | 456.45 | LA (1.1,5,7) | 1.10 | 722.622 | 7.81 | 10.87 |
| | Ti 2$p_{1/2}$ Ti Metal | 462.25 | LA (1.1,5,7) | 0.82 | 361.311 | | |
| | Ti 2$p_{3/2}$ Ti–N | 458.73 | GL (30) | 1.67 | 1143.81 | | 17.18 |
| | Ti 2$p_{1/2}$ Ti–N | 464.43 | GL (30) | 2.50 | 571.903 | | |
| O 1*s* | O$_{latt}$ | 530.44 | GL (30) | 1.50 | 3235.46 | | 14.02 |
| | OH, C–O, C=O, N–O | 532.10 | GL (30) | 1.50 | 15,499.1 | 2.93 | 67.06 |
| | O–C=O | 533.62 | GL (30) | 1.50 | 4378.49 | | 18.92 |
| N 1*s* | Al–N | 394.10 | GL (30) | 1.48 | 626.783 | | 47.33 |
| | Ti–O–N, Al–O–N | 396.16 | GL (30) | 1.48 | 262.838 | 1.8 | 19.82 |
| | Ti–N | 397.48 | GL (30) | 1.48 | 264.65 | | 19.93 |
| | N–O | 399.08 | GL (30) | 1.48 | 171.716 | | 12.92 |
| Al 2*p* | Al–N | 73.63 | LF (1,1,25,280) | 1.85 | 615.153 | 0.5371 | 65.35 |
| | Al–O–N | 76.05 | GL (30) | 2.12 | 326.468 | | 34.65 |

* Refers to the content of the element for a given oxidation state/binding form in the C 1*s* or K 2*p* region.

## 4. Conclusions and Discussion

As a result of research conducted to characterize the selected surface and structural properties of the titanium metallization coating deposited on the ceramics substrate by means of a friction surfacing process, the following conclusions can be made. The titanium coating has a thickness ranging from 3 to 7 μm. This value is large enough to achieve the continuity of the coating; at the same time, its limited thickness has a significant positive effect on the safe low state of residual stresses, lowering the risk of cracking the coating. The connection of the coating and the substrate is continuous; additionally, the coating's material effectively fills the unevenness of the surface of the substrate, caulking the structure of porous ceramics. The volume of the coating contains submicrometric particles of ceramic grains, which due to the friction of the metallization tool were detached from the substrate. Their even distribution indicates a high degree of plastic deformation of titanium during the formation of the coating. The stereoscopic structure of the surface layer of the metallization coating is anisotropic and is characterized by a relatively low roughness, with a mean value of $R_a$ = 0.404 μm in the y direction and 0.9 μm in the x direction. $R_z$, respectively 1.96 μm and 6.72 μm. The phase structure of the coating indicates its composite character, in which ceramic particles with dimensions significantly less than 1 μm are dispersed in the volume-dominant titanium matrix. The phase structure of the coating surface (XPS investigated) is dominated by TiN$_x$O$_y$ with the presence of TiO$_x$, TiN, metallic Ti, and AlN, additionally, carbon impurities and trace amounts of potassium were detected. Phase structure deeper below the surface (XRD investigated) is dominated by metallic Ti with AlN inclusions.

The peculiar properties of advanced ceramic materials and the resulting difficulties in bonding them with metals and/or the most challenging issue relate to the coating of advance ceramics with metals. Despite a large group of already developed methods and techniques of bonding ceramics with metals, obtaining ceramic–metal joints with high operational properties on the industrial scale will still be the subject of intense research [45–47]. Many new concepts for the production of ceramic–metal

joints are currently being developed, but due to the significant differences in the properties of the components, all the bonding methods are quite troublesome in mass production. The conducted experiments show the possibility of making a useful and economically attractive metallization titanium coating on the surface of AlN ceramics. Provided that argon gas protection is used, it is possible to eliminate the oxidation of titanium despite the heating to a relatively high temperature. The obtained coating is continuous, tight, and well-bonded to the substrate; its stereometric and phase structure favor the wettability of classic solders, e.g., AgCu28 in the context of using a metallization coating as a transition layer for soldering with metals.

In the case of fabricating the surface composites or metallic coatings, it is significantly difficult to avoid the interfacial reaction between the substrate, the coating metal matrix, and the formation of some detrimental phases and cracks. From a different point of view, the friction surfacing process is a solid-state surface modification method, which causes severe plastic deformation, a mixing of material, and lower (than melting point) temperature exposure, leading to significant microstructural refinement, densification, and homogeneity of the processed layer. Difficulties encountered in the fusion methods (particular regarding Ti coatings) for surface modification such as oxidation, cracks, porosity, segregation, and grain growth can be overcome by the friction surfacing process.

Friction surfacing has a significant potential for further industrial applications and is being developed as a practicable alternative to mainstream coating processes. This process allows the dissimilar joining of materials that would be otherwise incompatible or difficult to deposit by fusion-based methods. Friction surfacing has a significant potential for further industrial applications and is being developed as a practicable alternative to mainstream coating processes.

**Author Contributions:** Conceptualization, T.C. and M.H.; methodology, T.C.; software, M.H.; validation, T.C., M.H., T.S., and R.Ś.; formal analysis, M.H.; investigation, T.S., M.H., and R.Ś.; resources, M.H. and B.S.; writing—original draft preparation, T.C., M.H., A.K., and B.S.; writing—review and editing, T.C.; visualization, T.S., M.H.; supervision, T.C.; project administration, A.K. and B.S.; funding acquisition, T.C., T.S., and R.Ś.

**Funding:** This research was funded by using the statutory subsidy of the Faculty of Production Engineering of the Warsaw University of Technology in 2018.

**Acknowledgments:** The authors thank the deceased professor Władysław K. Włosiński for the inspiration to undertaken studies of solid-state process metallization of advanced ceramics.

**Conflicts of Interest:** The authors declare no conflict of interest.

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
