# Peer review of "Structure Investigation of Titanium Metallization Coating Deposited onto AlN Ceramics Substrate by Means of Friction Surfacing Process"

_coatings, doi:10.3390/coatings9120845_

Round 1
Reviewer 1 Report
I appreciate the authors' efforts in designing, conducting, and presenting this research. The research is interesting and has scientific merit, but the manuscript requires some revision before it can be published.
The introduction is not adequate neither in terms of explaining the concepts nor in terms of covering previous art. It would be better to mention some prominent applications of the target material in the introduction. Where, how, and why is it useful? Materials and methods is not adequately explained. Source of raw materials, mixture components and proportions of the ceramic should be described. The rotational friction technique should be described. The concepts and processes associated with this technique should be explained in light of the existing literature. Raw material characterization is better to be included in the materials and methods. Discussion of the results should be improved. In its current form, the discussion is more of a summary of results just mentioning main findings one by one. The discussion is expected to make sense of the results and relate the findings to each other in a logical manner. How is the durability of the coating? At minimum, the coating's durability under friction should be investigated or at least qualitatively discussed. The writing still needs some editing. There are a few writing problems, particularly with respect to word selection. For example, "friction tool" does not clearly reflect the described instrument's function. Please consider replacing it with a more representative name, unless it is a well-established application in the field (I've never ran into it though).Author Response
Dear reviewer,
Thank you very much for your effort and valuable advice leading to the improvement of the scientific quality of our manuscript.
Most of your comments have been taken into account, but for many reasons not fully. The main difficulty was quite divergent positions of as many as four reviewers.
Many of your valuable comments will be reflected in our future work.
with respect
authors
Reviewer 2 Report
Dear,
the overall article and the presented findings of your research are quite interesting, however, there are some issues and problems that, in my opinion, should be specified in more detail or changed.
Page 1: Abstract: It is known that the abstract should help readers remember key findings on a presented topic. The structure of the abstract can be as follows: 1) the research focus; 2) the research methods; 3) the results/findings of the research; 4) the main conclusions and recommendations. However, there are only the first two points in the presented abstract which makes it quite boring and tedious. I would recommend to re-write the abstract, e.g. line 15-16, is in my opinion absolutely unnecessary.
Line 21: Specify the chemical name of the compound - AlN. Abbreviations should be given following their explanations in the ‘Introduction’ section.
Line 24: Specify the term ‘Industry’ in more detail, there are very specific areas of the industry where this material can be used. Also, I would recommend to briefly specify the group of AlN ceramics.
Line 26: Bonding and welding are very different processes; therefore, I would recommend a change as follows: ‘Bonding ceramic materials with metals is one of the most difficult tasks in joining engineering.’
Chapter Introduction is very very weak. The discussion of the basic references related to the main topic does not exist. There are just numbers in brackets. This does not support the given statements, e.g. line 34: ‘In most cases of bonding ceramics with metals, during the joining process, the ceramics remain in the solid state [13-17],…’. What was investigated by reference no. 13, is it related to the topic of the presented article etc.
Line 38: ‘Currently, there is an interest in various types of surface modification processes and bonding with a mechanical method of energy supply [18-20].’ – Instead of the brackets, I would recommend saying:’ Currently, …. It was investigated by [18-20] that …’.
Line 38-41: It is difficult to understand the meaning of this sentence.
Chapter Material and Methods: It was stated that: ‘… the thermal expansion coefficient and the thermal conductivity coefficient are equally important for the properties of the formed joint.’ I assume that these material properties should be introduced before the investigation of the joint/bond. However, there isn't any information on the properties of the selected materials. Why?
Line 87-88: ‘Roughness parameters of the surface are on the relative low lever from used deposition method point of view.’ A reason or at least assumption for this result would be nice since it is not even in the discussion or conclusion section.
Line 96-97:’…, observations were carried out on the surface of the breakthrough obtained from the three-point bending of the sample, …’. Why you did not mention this test in the Method section. What was the test set up? What type of sample was used? What was the test speed, maximum load etc.
Line 109: It is said:’ in all types of coating’. What does it mean? Did you test more types of coatings?
Line 109-113: These sentences should be in Introduction.
Section Discussion – As it is stated in line 189: ‘… the following conclusions can be made.’, this is not discussion. The findings and results are discussed in the previous section, therefore, I would recommend re-writing both chapters or rename the sections according to their content.
Section Conclusion: Merge this section with the content of the discussion.
Line 207-208: ‘The peculiar properties of ceramic materials and the resulting difficulties in bonding them with metals are an important technological problem.’ I do not agree with this statement. Bonding of ceramics and metals is not a big deal, the problem is the bonding of advanced ceramics and metals AND/OR the most challenging issue is the coating of advance ceramics!!!
Author Response
Dear reviewer,
Thank you very much for your effort and valuable advice leading to the improvement of the scientific quality of our manuscript.
Most of your comments have been taken into account, but for many reasons not fully. The main difficulty was quite divergent positions of as many as four reviewers.
Many of your valuable comments will be reflected in our future work.
with respect
authors
Our answers are below:
Page 1: Abstract: It is known that the abstract should help readers remember key findings on a presented topic. The structure of the abstract can be as follows: 1) the research focus; 2) the research methods; 3) the results/findings of the research; 4) the main conclusions and recommendations. However, there are only the first two points in the presented abstract which makes it quite boring and tedious. I would recommend to re-write the abstract, e.g. line 15-16, is in my opinion absolutely unnecessary.
The corrections were made in accordance with the reviewer's comment.
Line 21: Specify the chemical name of the compound - AlN. Abbreviations should be given following their explanations in the ‘Introduction’ section.
The corrections were made in accordance with the reviewer's comment.
Line 24: Specify the term ‘Industry’ in more detail, there are very specific areas of the industry where this material can be used. Also, I would recommend to briefly specify the group of AlN ceramics.
The corrections were made in accordance with the reviewer's comment.
Line 26: Bonding and welding are very different processes; therefore, I would recommend a change as follows: ‘Bonding ceramic materials with metals is one of the most difficult tasks in joining engineering.’
The corrections were made in accordance with the reviewer's comment.
Chapter Introduction is very very weak. The discussion of the basic referencesrelated to the main topic does not exist. There are just numbers in brackets. This does not support the given statements, e.g. line 34: ‘In most cases of bonding ceramics with metals, during the joining process, the ceramics remain in the solid state [13-17],…’. What was investigated by reference no. 13, is it related to the topic of the presented article etc.
The corrections were made in accordance with the reviewer's comment.
Line 38: ‘Currently, there is an interest in various types of surface modification processes and bonding with a mechanical method of energy supply [18-20].’ – Instead of the brackets, I would recommend saying:’ Currently, …. It was investigated by [18-20] that …’.
The corrections were made in accordance with the reviewer's comment.
Line 38-41: It is difficult to understand the meaning of this sentence.
The corrections were made in accordance with the reviewer's comment.
Chapter Material and Methods: It was stated that: ‘… the thermal expansion coefficient and the thermal conductivity coefficient are equally important for the properties of the formed joint.’ I assume that these material properties should be introduced before the investigation of the joint/bond. However, there isn't any information on the properties of the selected materials. Why?
The corrections were made in accordance with the reviewer's comment.
Line 87-88: ‘Roughness parameters of the surface are on the relative low lever from used deposition method point of view.’ A reason or at least assumption for this result would be nice since it is not even in the discussion or conclusion section.
The corrections were made in accordance with the reviewer's comment.
Line 96-97:’…, observations were carried out on the surface of the breakthrough obtained from the three-point bending of the sample, …’. Why you did not mention this test in the Method section. What was the test set up? What type of sample was used? What was the test speed, maximum load etc.
The corrections were made in accordance with the reviewer's comment.
Line 109: It is said:’ in all types of coating’. What does it mean? Did you test more types of coatings?
The corrections were made in accordance with the reviewer's comment.
Line 109-113: These sentences should be in Introduction.
The corrections were made in accordance with the reviewer's comment.
Section Discussion – As it is stated in line 189: ‘… the following conclusions can be made.’, this is not discussion. The findings and results are discussed in the previous section, therefore, I would recommend re-writing both chapters or rename the sections according to their content.
The corrections were made in accordance with the reviewer's comment.
Section Conclusion: Merge this section with the content of the discussion.
The corrections were made in accordance with the reviewer's comment.
Line 207-208: ‘The peculiar properties of ceramic materials and the resulting difficulties in bonding them with metals are an important technological problem.’ I do not agree with this statement. Bonding of ceramics and metals is not a big deal, the problem is the bonding of advanced ceramics and metals AND/OR the most challenging issue is the coating of advance ceramics!!!
The corrections were made in accordance with the reviewer's comment.
Reviewer 3 Report
Please see the attached file.

Author Response
Dear reviewer,
Thank you very much for your effort and valuable advice leading to the improvement of the scientific quality of our manuscript.
Most of your comments have been taken into account, but for many reasons not fully. The main difficulty was quite divergent positions of as many as four reviewers.
Many of your valuable comments will be reflected in our future work.
with respect
authors
Reviewer 4 Report
Describe all methods used for characterization into the section Materials and Methods
At point 3.2. authors mention that “Figure 3a shows the microstructure of the ceramic substrate joint (AlN) with a metallic coating 95 (Ti).” but the figure is just an image with morphological aspects of the joint (probably cross-section). The image didn’t show the microstructure.
The authors mention “the three-point bending of the sample” but didn’t present any details about the this test.
I suggest completing the figure 3 and 4 with some images showing the aspects about metallization coating surface (SEM, IR microscopy or AFM).
I suggest completing the figure 5 with some images showing the aspects about the interface between the ceramic substrate joint (AlN) and the metallic coating (Ti). The Ti/AlN interface must be shown also at the lower magnification, in order to see the entire profile of the joint.
Also, the authors could add an image of the fracture of a Ti coating/AlN substrate system. The fracturing of the sample, the authors could make a severe test of the adherence of the metallic coating to the ceramic substrate.
“AlN grains were also present in the coating structure” – please provide some comments about this fact.
“Large amounts of carbon impurities and trace amounts of potassium were detected on the surface of deposited Ti coating” – please provide some comments about this fact.
I didn’t agree with the sentence “The presence of Al metal cannot be excluded.”(line 185). Please provide some comments or made some new investigations in order to prove this sentence.
Some grammar mistakes must be corrected by the authors. (e.g. “relative low lever”, “cross section thru”, “Figure 3b shows the SEM image”).
Author Response
Dear reviewer,
Thank you very much for your effort and valuable advice leading to the improvement of the scientific quality of our manuscript.
Most of your comments have been taken into account, but for many reasons not fully. The main difficulty was quite divergent positions of as many as four reviewers.
Many of your valuable comments will be reflected in our future work.
with respect
authors
Our Answers are below:
Describe all methods used for characterization into the section Materials and Methods
The corrections were made in accordance with the reviewer's comment.
At point 3.2. authors mention that “Figure 3a shows the microstructure of the ceramic substrate joint (AlN) with a metallic coating 95 (Ti).” but the figure is just an image with morphological aspects of the joint (probably cross-section). The image didn’t show the microstructure.
The corrections were made in accordance with the reviewer's comment.
The authors mention “the three-point bending of the sample” but didn’t present any details about the this test.
The corrections were made in accordance with the reviewer's comment.
I suggest completing the figure 3 and 4 with some images showing the aspects about metallization coating surface (SEM, IR microscopy or AFM).
The corrections were made in accordance with the reviewer's comment.
I suggest completing the figure 5 with some images showing the aspects about the interface between the ceramic substrate joint (AlN) and the metallic coating (Ti). The Ti/AlN interface must be shown also at the lower magnification, in order to see the entire profile of the joint.
The purpose of the article is not to analyze the interface but to study the structure of the coating. We will devote another article to the interface structure.
Also, the authors could add an image of the fracture of a Ti coating/AlN substrate system. The fracturing of the sample, the authors could make a severe test of the adherence of the metallic coating to the ceramic substrate.
We will devote another article to the interface structure.
“AlN grains were also present in the coating structure” – please provide some comments about this fact.
This sentence has been deleted. Comment regarding of this problem has been located in conclusions
“Large amounts of carbon impurities and trace amounts of potassium were detected on the surface of deposited Ti coating” – please provide some comments about this fact.
The correction was made. Present state is "Low amounts" as should be early according to the real state.
I didn’t agree with the sentence “The presence of Al metal cannot be excluded.”(line 185). Please provide some comments or made some new investigations in order to prove this sentence.
This sentence has been deleted.
Some grammar mistakes must be corrected by the authors. (e.g. “relative low lever”, “cross section thru”, “Figure 3b shows the SEM image”).
The corrections were made in accordance with the reviewer's comment.
Round 2
Reviewer 1 Report
For future record, please also submit a clear version of the revised manuscript, and justify all changes and revisions in a separate document. The reviewer comments should be responded one by one in that document.
the authors have promised to address the comments in their future work, which I do not find reasonable. The subject here is the present manuscript which should have sufficient merit for being published.
The research presented in this manuscript has acceptable scientific merit and is of interest for the reader. I commend the authors for conducting this research. However, the submitted manuscript poorly represents the conducted research, and it has not been substantially improved during revision.
Reviewer 2 Report
The paper was corrected and in my opinion, can be accepted in its present form.